# Supercritical Carbon Dioxide Antisolvent Fractionation for the Sustainable Concentration of *Lavandula luisieri* (Rozeira) Riv.- Mart Antimicrobial and Antioxidant Compounds and Comparison with Its Conventional Extracts

**DOI:** 10.3390/plants8110455

**Published:** 2019-10-26

**Authors:** Carlota Giménez-Rota, Susana Lorán, Ana M. Mainar, María J. Hernáiz, Carmen Rota

**Affiliations:** 1GATHERS Group, Aragón Institute of Engineering Research (I3A), University of Zaragoza, c/. Mariano Esquillor s/n, 50018 Zaragoza, Spain; carlotagimenezrota@gmail.com (C.G.-R.); ammainar@unizar.es (A.M.M.); 2Chemistry in Pharmaceutical Science Department, Pharmacy Faculty, Complutense University of Madrid, Plaza Ramón y Cajal s/n, 28040 Madrid, Spain; mjhernai@farm.ucm.es; 3Department of Animal Production and Food Science, AgriFood Institute of Aragon (IA2), University of Zaragoza-CITA, Veterinary Faculty, University of Zaragoza, Miguel Servet 177, 50013 Zaragoza, Spain; crota@unizar.es

**Keywords:** SAF, antioxidant activity, antimicrobial activity, ursolic acid, oleanolic acid, rosmarinic acid

## Abstract

*Lavandula stoechas* subsp. *luisieri* is a Spanish subspecies from the *Lamiaceae* family. Its essential oil has been traditionally used for several medical applications though little is known about other extracts. Similar to many other studies aiming to obtain traditional plant extracts to be used in different applications, this work evaluated the antioxidant and antimicrobial activities of *Lavandula luisieri* extracts and the correlation with their composition. Traditional hydrodistillation and ethanolic maceration were used to obtain the essential oil and the maceration extract, respectively. A green and sustainable methodology was applied to the maceration extract that was under a Supercritical Antisolvent Fractionation process to obtain a fine solid enriched in rosmarinic acid and the terpenes oleanolic and ursolic acids. Antimicrobial activities of all extracts and pure identified compounds (rosmarinic and ursolic acids) were evaluated against five bacterial strains; *Listeria monocytogenes*, *Enterococcus faecium*, *Staphylococcus aureus*, *Salmonella* Typhimurium and *Escherichia coli* and were compared with the pure compounds identified, rosmarinic and ursolic acids. All strains were sensitive against *L. luisieri* essential oil. The solid product obtained from the supercritical process was concentrated in the identified actives compared to the maceration extract, which resulted in higher antimicrobial and DPPH scavenging activities. The supercritical sustainable process provided *L. luisieri* compounds, with retention of their antimicrobial and antioxidant activities, in a powder exemptof organic solvents with potential application in the clinical, food or cosmetic fields.

## 1. Introduction

Plants have been used since ancient times for their perfume, flavor, and preservative properties in a variety of products and applications with medicinal and cosmetic uses [1] because of their secondary metabolites with diverse bioactivities [2]. Nowadays, population concern about healthier and more natural habits has promoted the study of plant extracts in different research fields, specifically to find new molecules for various applications. Indeed, it is estimated that over a hundred of new natural product-based leads are under clinical development [3] to prevent and treat chronic illnesses, whose physiopathology is based on oxidative stress, such as cardiovascular problems, diabetes or Alzheimer disease [4]. This same trend is followed by the cosmetic, food or agricultural industries, where the substitution of synthetic chemical additives or pesticides for natural ones has been demanded [5].

The Spanish lavender or *Lavandula luisieri* (Rozeira) Riv. Mart. is an aromatic *Lamiaceae* widespread in the southwest of the Iberian Peninsula. It has been traditionally used as an antiseptic of for wounds, antiaging for mature skin and scar healer, or as an antispasmodic and digestive. Its study has been centered on its essential oil, because of the atypical compounds that produces. Besides 1,8-cineole, lavandulol, linalool and their acetates, also found in other *Lavandula* species, *L. luisieri* possesses a series of compounds with a 1,2,2,3,4-pentamethylcyclopentane (necrodane) structure [6,7,8,9]. These constituents have only been previously found in the defensive secretions of the beetle *Necrodes surinamensis* and in the sexual pheromone of the grape mealybug *Pseudococcus maritimus* [10]. The potential bioactivity of this particular plant species has been focused so far on its essential oil, for which several bioactivities have been reported; antioxidant [11,12,13]; antimicrobial [6,13,14,15,16,17] antifeedant [7,8]; ixodicidal [18]; antiparasitic [19]; and antiinflamatory [13,20]. However, information about other *L. luiseri* extracts is scarce. So far, only two other studies have obtained its extract but a high temperature was applied [8,21]. Nunes et al., [21], reported a high content in polyphenols such as rosmarinic, chlorogenic, and ferulic acids in *L. luiseri* ethanolic extracts. Julio et al., [8] also identified terpene compounds such as oleanolic ursolic and tormentic acids. Rosmarinic acid, the most abundant compound in both studies, is a dimer of caffeic acid, possesses potential biological activities such as antiviral, antibacterial, anti-inflammatory and antioxidant activities [22]. In addition, oleanolic and ursolic acids, two isomer triterpenoid compounds, have hepatoprotective, anti-inflammatory, antimicrobial and anticancer effects. [23,24].

Plant extracts are usually obtained by traditional techniques, operating at high temperatures, as in hydrodistillation to obtain the essential oil or soxhlet to obtain its polar compounds. These methods can cause either the degradation of actives from plant material because of the harsh conditions and environmental contamination or toxicity as well as the use and traces of organic solvent. The design of green and sustainable processes is currently a hot research topic in food, cosmetic, agronomic and pharmaceutical industries. In this regard, supercritical CO_2_ (scCO_2_) is a suitable extractor to obtain natural actives from vegetable sources [25,26] since its supercritical conditions, 72 bar and 32 °C, are mild enough to avoid their degradation. In addition, CO_2_ is harmless and it can be easily separated from the extract by a simple change of pressure, which generates a final product without residual presence of this solvent [27]. ScCO_2_ has been studied in research and applied at large scale by the industry for the extraction of natural compounds from plant matrices. Its adjustable solvation power can be applied in the separation and purification of certain compounds from complex mixtures [28]. Supercritical antisolvent fractionation (SAF) with CO_2_ has been reported to be a promising technique to reach this purpose. It allows the concentration of actives from an organic solution at mild conditions and precipitates them into solid particles with controlled shape and diameter without damaging organic solvents [29]. In SAF, a solution of the organic extract is continuously pumped and sprayed into a vessel where it converges with scCO_2_. The fraction of compounds insoluble in this new mixture of solvent-scCO_2_ precipitate, while the remaining portion, along with the solvent, is collected in the second vessel as a solution. This technique has been applied to obtain enriched fractions that could enhance certain bioactivities [30,31,32,33]. 

The aim of this work was the determination of the antimicrobial and antioxidant activities of different extracts, essential oil, maceration extract and SAF fractions of *L. luisieri*, and their correlation with their composition.

## 2. Materials and Methods

### 2.1. Reagents

Solvents used were hexane (99.0% Panreac, Barcelona, Spain), ethanol (EtOH, 99.8%, Sigma Aldrich, Madrid, Spain), Carbon dioxide (CO2, 99.8% ALPHA GAZ, Madrid, Spain) and dimethyl sulfoxide (DMSO, SigmaAldrich Química, Madrid, Spain). For the chromatographic analysis mobile phase, the following solvents were used: water ultrapure 18.2 MΩ·cm filtered through 0.2 µm (Milli-Q-Plus apparatus from Millipore, mod Milford, MA), phosphoric acid (85.9% Fluka), methanol and acetonitrile (MetOH and ACN, 99.9% Scharlab, Barcelona, Spain). Chromatographic standards, rosmarinic oleanolic and ursolic acids (99% 99% and 99.8%) were purchased from Sigma-Aldrich, Madrid, Spain. For the antioxidant activity assay, we used the free radical 2,2-diphenyl-1-picrylhydrazyl (DPPH, Sigma Aldrich, Madrid, Spain) and as a positive control, 6-hydroxy-2,5,7,8-tetramethylchromane-2-carboxylic acid (Trolox, 97% ACROS Organics, Madrid, Spain).

### 2.2. Plant Material

*Lavandula luisieri* plant material was collected from Cariñena, Zaragoza (Spain) and provided by the Agrifood Research and Technology Centre of Aragon (CITA, Spain). This plant was adapted to the Cariñena experimental field in 2008 from a Toledo (Spain) wild population. It was cultivated and collected as reported by Julio et al., [34]. Plant material was dried at room temperature, pulverised and sieved. Its mean diameter was adjusted to 0.330 mm (ASAEA S319.3 from the American National Standards Institute).

### 2.3. Lavandula luisieri Extracts 

#### 2.3.1. Essential Oil

The essential oil was obtained by hydrodistillation in a Clevenger-type apparatus from *Lavandula luisieri* previously described. The essential oil was provided by the Agrifood Research and Technology Centre of Aragón (CITA), and preserved in an amber vial under refrigeration. Its chemical composition has been reported in a previous study [18].

#### 2.3.2. Maceration Extract

The polar fraction of *L. luisieri* was obtained by submitting the pulverized plant material to two serial macerations: first with hexane, to eliminate the apolar compounds such as cuticular wax, and second with ethanol, to obtain the polar and active compounds. Macerations were carried out at room temperature for 48 hours, with a plant material:solvent ratio of 1:10 (w/w). Both solvents were removed with a rotary evaporator (Büchi R-200) to obtain the dry extract. Only the ethanolic extract (ME) was considered to perform the tests. 

The yield of the maceration extract was determined according to Equation (1).
(1)$$Y_{ME}\left(wt.\%\right)=\left(\frac{mass{\;}_{plant\;extract}\left(g\right)}{mass{\;}_{plant\;material}\left(g\right)}\right)\times 100$$
where mass_plant extract_ is the mass of the extract obtained (dry extract, with the solvent removed), and mass_plant material_ was the initial mass of dried and pulverised plant submitted to the extraction process. 

#### 2.3.3. Supercritical Antisolvent Fractionation

Part of the ethanolic maceration extract was dissolved in ethanol 3% (w/w), and filtered through a 0.45 μm pore size to constitute the feed solution (FS) of the SAF process. SAF was performed using a laboratory-scale apparatus (Green Chemistry Laboratory, I3A Researching Institute at University of Zaragoza) equipped with a CO_2_ pump, an extract solution pump, a 0.5 L precipitation vessel and a downstream collector as main components (Figure 1).

The FS was pumped towards the precipitation vessel with a flow rate of 0.5 mL/min. The SC-CO_2_ flow rate was 30 g/min and the final pressure was 130 bar, i.e., conditions selected for a higher mass recovery according to previous SAF optimization (Figure 1). The insoluble compounds in the mixture ethanol-supercritical CO_2_ precipitate in this vessel constitute the precipitation vessel fraction (PV). Those compounds that were still soluble in the mixture were collected, along with the solvent ethanol, as the downstream vessel fraction (DV). The mass recovery yields in PV and DV were determined according to Equation (2).
(2)$$Y_{i}\left(wt\%\right)=\left(\frac{mass\;fraction\;collected_{i}}{mass\;of\;FS}\right)\times 100$$
where i is the location of PV or DV collection.

### 2.4. Chemical Composition Analysis

*Lavandula luisieri* maceration extract and its supercritical fractions were analysed with the following chromatographic procedure. The equipment used was an High Performance Liquid Chromatography (HPLC) Waters^®^ Alliance 2695 with a PDA Waters^®^ 2998 detector provided with the column CORTECS^®^ C18 2.7 μm (4.6 × 150 mm) and a CORTECS^®^ Pre-column VanGuard C18 2.7 μm (2.1 × 5 mm) (Barcelona, Spain). For the chromatographic analysis, an isocratic mobile phase of 88:12, methanol (MeOH): H_3_PO_4_ 0.5% in Milli-Q water, was pumped for 10 min at 0.8 mL/min. Extracts were dissolved in ethanol to 100 ppm (approximately), filtered through a GH Polypropylene membrane (ACRODISC 13 mm, 0.2 μm, Waters, Barcelona, Spain) and finally 10 µL was injected. In order to quantify the maceration extract, PV and DV the identified rosmarinic, oleanolic and ursolic acids, standards were used to build the calibration curves. Rosmarinic acid was measured at 330 nm, whereas ursolic and oleanolic acids were measured at 210 nm. All analyses were performed in triplicate. The actives content was expressed as a percentage of the dry extract or SAF fraction analyzed. 

### 2.5. Antimicrobial Activity Assays

#### 2.5.1. Microorganisms and Growth Conditions

The bacterial strains assayed in this study were obtained from the Spanish Collection of Type Cultures (CECT) included and maintained frozen at −80 °C in cryovials until the sensitivity tests. Three gram-positive bacteria, *Listeria monocytogenes* (CECT 911), *Enterococcus faecium* (CECT 410) and *Staphylococcus aureus* (CECT 435), and two gram-negative bacteria, *Salmonella* Typhimurium (CECT 443) and *Escherichia coli* (CECT 516), were selected for the assays.

Broth subcultures were prepared by inoculating, with one single colony from a Tryptic Soy Agar (TSA, Oxoid, Madrid, Spain) plate, a test tube containing 10 mL of sterile Tryptic Soy Broth (TSB, Oxoid, Madrid, Spain). The inoculated tubes were incubated overnight (16 hours) at 37 °C. Then, the bacterial concentration was adjusted to an absorbance between 0.08 to 0.1 using a spectrophotometer (Jenway 3600, Tirana, Albania) with a wavelength of 620 nm which corresponds to 1 × 10^8^ UFC/mL according to McFarland Turbidity scale (Standart N1 0.5, Becton Dickinson and Company, Madrid, Spain). Additionally, inocula concentration was confirmed by colony counting in agar plates after performing 1:10 dilutions in peptone water (Buffered peptone water, Oxoid, Madrid, Spain).

#### 2.5.2. Disk Diffusion Method

Antimicrobial activity of the essential oil was tested against the five bacterial strains using the agar disk diffusion technique. Filter paper disks (Whatman 6 mm diameter, Sigma- Aldrich, Madrid, Spain) containing 15 µL of essential oil were placed on the surface of agar plates of Mueller-Hinton (Merck, Spain) that were previously seeded by spreading one sterile swab impregnated in strain culture of 1 × 10^8^ UFC/mL. Ampicillin disks (10 µg, Oxoid, Madrid, Spain) were used as a positive control. The plates were incubated at 37 °C for 24 hours, and the diameter resulting from each inhibition zone (diameter of inhibition zone plus diameter of the disk) was measured in triplicate in millimeters. The scale of measurement was the following: ≥ 20 mm is strongly inhibitory, < 20-12 mm moderately inhibitory, and < 12 mm is non-inhibitory [35]. An average and standard deviation of the inhibition zone from three replicates were calculated.

#### 2.5.3. Determination of Minimum Inhibitory Concentration (MIC) and Minimum Bactericide Concentration (MBC)

Antimicrobial activity of *Lavandula luisieri* extracts (essential oil, maceration extract, PV and DV) was quantified against the five working strains by the determination of the minimum inhibitory concentration (MIC) and the minimum bactericidal concentration (MBC). These values were assessed for *Lavandula luisieri* essential oil by the macrodilution broth technique, whereas for ME, PV and DV, the microdilution broth technique was applied. The final bacterial working suspension for both assays was adjusted to 5 × 10^5^–1 × 10^6^ CFU/mL by dilution from the measured 10^8^ UFC/mL overnight culture. For both procedures, macrodilution and microdilution, the MIC was the lowest concentration of extract at which bacteria failed to grow, so no visible changes were detected in the broth medium, and the MBC was defined as the concentration at which bacteria were reduced by 99.9%.

The essential oil activity was tested using the macrodilution method adapted from Clinical and Laboratory Standards Institute (CLSI, M07-A10, 2018). The assays were performed in 10 mL of TSB (ethanol 3%), and the tested concentrations were obtained by adding suitable amounts of essential oil to a final working range of 0.5–30 μL/mL. Positive controls contained TSB with microorganisms plus 3% ethanol. Negative controls contained TSB plus 3% ethanol and 5 μL/mL of *Satureja montana* essential oil, whose activity has been widely studied and proved [35,36]. After a 24 h incubation at 37 °C in a shaking thermostatic bath (Bunsen, mod. BTG), the MIC was read as the concentration with no visible growth. In order to evaluate MBC, 100 μL of each case in which microbial growth was not observed was spread plated in TSA. Plates were incubated at 37 °C for 24 h. The evaluation of MIC and MBC values was carried out in triplicate.

*Lavandula luisieri* ME, PV and DV extracts were tested against the same bacterial strains with the microdilution broth method [37]. The test was performed in 96-well sterile microplates. All wells received Mueller Hinton Broth (MHB) supplemented with 10% glucose and 1% phenol red broth (Merck, Madrid, Spain). Extract working solutions were dissolved in water with DMSO 5% with a final well highest concentration of 2.5% [38]. The solutions were sterilized by filtration with a 0.2 μm pore membrane filter and added to the first column of wells in the microplate. The final extract concentration assayed ranged from 2000 to 2 μg/mL and was obtained by twofold serial dilution from the first column.

Finally, inoculum suspension was added to all wells. The growth controls constituted medium with extract (negative control) and medium with bacterial inoculum (positive control).

Each microplate was incubated for 24 h at 37 °C. A change of color from red to yellow was interpreted as positive growth. For MBC determination, 10 μL from each well presenting no visible growth was inoculated on Mueller Hinton agar plates and incubated at 37 °C for 24 h. Each analysis was performed in triplicate.

The standards rosmarinic and ursolic acids were also tested following the same microdilution procedure with final concentrations of 125 to 0.06 μg/mL and 1000 to 0.5 μg/mL, respectively, according to its proportion in the *L. luisieri* studied extracts. For the results of antimicrobial activity, the following were considered: significantly active when MIC < 100 μg/mL, moderately active 100 < MIC > 625 μg/mL and weakly active MIC > 625 μg/mL [37].

### 2.6. Antioxidant Activity 

The capacity of the *L. luisieri* maceration extract and its supercritical fractions to scavenge DPPH free radicals was measured by an adaptation of the Brand-Williams, Cuvelier and Berset (1995) [39] spectrophotometric method. The extract solutions were mixed 1:1 (v/v) with a DPPH ethanol solution of 40 µg/mL. The DPPH solution was also confronted with pure RA and trolox (97% ACROS Organics) as a positive control and with ethanol as a negative control. The final well concentrations of ME, PV, DV and the positive controls rosmarinic acid and trolox ranged from 300 - 0.1 µg/mL. The absorbance was measured at 520 nm after 30 min of reaction at room temperature with a microplate photometer (Multiskan EX mod. 355, Thermo Labsystems, Zaragoza, Spain). To determine the scavenging capacity the following equation was applied, Equation (3):(3)$$Radical\;scavenging\;activity\;\left(wt\%\right)=\left[\left(Abs_{control}-Abs_{sample}\right)/\left(Abs_{control}\right)\right]\times 100$$
where Abs_control_ is the measured absorbance of the DPPH solution and Abs_sample_ is the measured absorbance after the reaction between the extracts or control vs. DPPH. 

The antioxidant activity of plant extracts was expressed as IC_50_, which is defined as the concentration of extracts (in µg/mL) required to inhibit 50% of DPPH radicals. IC_50_ values were estimated by a nonlinear regression (GraphPad Prism version 4.0). A lower IC_50_ value indicates higher antioxidant activity. The results are given as the mean ± standard deviation (SD) of experiments performed in triplicate.

## 3. Results and Discussion

### 3.1. Chemical Composition of Lavandula luisieri Extracts 

The non-volatile and polar fraction of this plant was obtained by ethanolic maceration. The final extraction yield obtained with this maceration was 12.2%. Regarding this same *Lavandula luisieri* population, other authors [18] obtained similar yield extraction (12.5%), although it was obtained with ethanol in a Soxhlet apparatus. This method applies heat for a long time and enhances the extraction of actives but it also can cause the degradation of the thermolabile compounds; it does not seem to provide better extraction yield. In this work, the extraction was performed at room temperature and in amber bottles to prevent degradation from heat and light.

The ethanolic maceration extract, ME, was fractionated through SAF into two fractions.It was recovered 50.9% (w/w) in the PV fraction and a 32.9% (w/w) in the DV fraction. According to these results, a high quantity of compounds was insoluble in the mixture scCO_2_-EtOH under these experimental conditions of pressure and CO_2_ flow rate, since a 50.9% of the initial mass precipitated when the ethanolic solution encountered the scCO_2_ in the PV. 

The composition of ME, PV and DV was studied by means of HPLC and only the polyphenol rosmarinic acid and the two triterpenes ursolic acid and oleanolic acid were identified (Figure 2). The content of these actives was previously reported [8]. The concentration of rosmarinic, oleanolic and ursolic acids was quantified in each fraction and compared to the initial ME (Table 1).

The rosmarinic acid concentration was 51.4 mg/g of the initial maceration extract. After the supercritical fractionation process, rosmarinic acid was entirely retained in the PV fraction, at a final concentration of 93.5 mg/g of PV. Although ursolic acid and oleanolic acid were distributed between both fractions, ursolic acid also precipitated mostly in the first fraction, reaching a concentration of 118.6 mg/g of PV; meanwhile, oleanolic acid seem to be dragged to the downstream vessel. The total quantity of the studied compounds in the PV was 238.5 mg/g, 1.9 times more concentrated compared to the initial ME.

Other authors have previously performed this process of supercritical concentration for natural compounds. The application of SAF to rosemary extracts in order to concentrate its polyphenols, rosmarinic acid among them, produced a fraction with a higher antiproliferative and antioxidant activity. Sánchez-Camargo et al., [32] and Visentin et al., [33]. Bernatoniene et al., [40] also studied different techniques for the extraction of these three active constituents from *Rosmarinus officinalis*. The highest mass of active per gram of extract was achieved with ultrasound-assisted ethanolic extraction for ursolic acid (15.8 ± 0.2 mg/g) and rosmarinic acid (15.4 ± 0.1 mg/g), and with ethanolic maceration for oleanolic acid (12.2 ± 0.1 mg/g).

In our work, the ethanolic extraction revealed that *L. luisieri* is a rich source of RA, OA and UA, and the supercritical antisolvent fractionation process is a useful technique for the co-precipitation of the followed actives into a solid enriched product, with 93.5 mg/g, 118.5 mg/g and 26.4 mg/g of RA, OA and OA, respectively.

### 3.2. Essential Oil Antimicrobial Activity

Bacterial susceptibility of *L. luisiseri* essential oil was first tested with the paper disk agar diffusion method and then quantified with the broth dilution method. The results indicated in Table 2 represent bacterial sensitivity according to the classification previously described [35].

*E. faecium* (10.5 ± 0.3 mm) and *E. coli* (11.5 ± 0.2) showed no susceptibility to *L. luisieri* essential oil. *L. monocytogenes* was the most susceptible bacteria (43.3 ± 2.8 mm), followed by *S. aureus* (36.2 ± 2.9 mm) and finally *S.* Typhimurium (21.4 ±0.4 mm). With the agar disk diffusion technique, qualitative information of *L. luisieri* essential oil antimicrobial activity was obtained. 

Regarding MIC and MBC values, *L. luisieri* essential oil had bacteriostatic activity against all bacteria tested, showing a wide antibacterial spectrum inhibiting both, gram-positive and gram-negative bacteria (Table 2). The highest bacteriostatic activity obtained was against gram-positive tested (MIC = 0.5 μL/mL), whereas higher MIC values were acquired against gram-negative ones: 5 μL essential oil /mL against *S.* Typhimurium and 30 μL essential oil /mL vs *E. coli*. The evaluation of MBC revealed that *L. luisieri* essential oil was bactericidal against four of the five bacteria at the assayed concentrations. The bactericidal MBC values ranged from 0.5 μL/mL (*S. aureus*) to 5μL/mL (*S.* Typhimurium). *E. coli* seems to be the most resistant strain to this oil composition, since it showed bacteriostatic but not bactericidal activity, at the highest concentration tested, 30 μL. This behavior was also observed by Baldovini et al., [6], by applying a dilution agar method with a chemically defined necrodyl-rich essential oil from *L. luisieri* against different bacterial and yeast strains. Although the activity was tested with a different method, it was also observed that gram-positive strains were more sensitive to the essential oil than the gram-negative ones. Some *E. coli* strains had sensitivity to *L. luisieri* essential oil, but also at the highest concentration tested.

*L. luisieri* essential oil has a higher antimicrobial activity compared to other *Lavandula* species, which only had activity against *S. aureus* such as *L. lavandulifolia*, *L. angustifolia*, Lavandin Super and Lavandin Abrialis [41]. Nevertheless, other species such as *L. latifolia* and Lavandin Grosso had similar MIC and MBC results to *L. luisieri* against several strains such as *S.* Typhimurium, *E coli* and *L. monocytogenes*. Generally, *Lavandula* species essential oil, as other plant families, has different sensitivity among gram-positive and gram-negative strains, with the gram-negative *E. coli* as the most resistant one. Nevertheless, the chemical composition of this other antimicrobial species differs with *L. luisieri*, and the activity was related to their content in 1,8-cineol, linalool and camphor [42].

So far, only Baldovini et al. [1] Roller et al. [16] and Zuzarte et al. [17] have tested *L. luisieri* essential oil antimicrobial activity. They obtained good results against all strains tested, especially against gram-positive bacterial and fungal strains, such as *Candida albicans* and *Aspergillus* species, and related these results to *L. luisieiri* essential oil necrodane terpenoid content. Roller, Ernest and Buckle [16] compared *Lavandula* species antimicrobial activity against methicillin resistant and sensitive *Staphylococcus aureus* with the disk diffusion method. *L. luisieri* essential oil, highly concentrated in necrodane monoterpenoids, with 34.5% relative content in α-necrodyl acetate, was the most active at lower volumes. In addition, its combination (50:50) with *L. stoechas* and *L. angustifolia* essential oils produced inhibitory zones twice the diameter than the zones obtained with both single oils. A possible synergy among necrodane terpenoids and 8-cineole, fenchone, and camphor from *L. stoechas*, or linalool and linalyl acetate from *L. angustifolia*, is suggested. Similarly, Zuzarte et al., [17] also related *L. luisieri* essential oil antimicrobial activity to its particular content in necrodane-type compounds when studied in two different populations from central and southern regions of Portugal against different *Aspergillus* strains. The activity results were considered relevant since this strain is usually less sensitive to essential oils. 

In our work, the obtained essential oil also had a good antimicrobial activity. Its composition was reported in a previous study [18]. Its main compounds were camphor (60.3%) and 2,3,4,4-tetramethyl-5-methylidenecyclopent-2-en-1-one (8.5%), a necrodane-type compound, along with other substances such as fenchone (2.9%) and 1,8-cineol (2.0%). Other authors studying *L. luisieri* essential oil composition observed that the differences among populations appeared to be only quantitative [6,8,9,18,43]. The antimicrobial activity results obtained in this work could be a consequence of *L. luisieri* essential oil high content in camphor, which is a well-known antimicrobial compound, obtained mainly from *Cinnamomum camphora* but widely distributed in the plant kingdom [44]. According to other authors’ results, besides camphor, the *L. luisieri* content in atypical necrodane compounds and the combination with other volatiles could also be contributing to this activity.

### 3.3. Antimicrobial Activity of SAF Fractions 

The antimicrobial activity of the ethanolic extract and its supercritical fractions PV and DV was tested. To our knowledge, this is the first work to study the antimicrobial activity of polar non-volatile fractions of this plant. For this type of extract, the disk diffusion method could not be applied as a screening method because of the poor diffusion from the cellulose disks to the agar. The antimicrobial activity of ME, PV and DV, as well as the pure compounds rosmarinic and ursolic acids, was tested and quantified with the microdilution broth procedure. From all studied strains assayed, only *L. monocytogenes*, *E. faecium* and *S. aureus* showed sensitivity to the *L. luisieri* maceration extract and its supercritical fractions. However, *Salmonella* Typhimurium and *Escherichia coli* did not show sensitivity to the extracts at the assayed concentrations (Table 3).

Rosmarinic acid did not show any antimicrobial activity in the studied range (0.06–125 μg/mL). Although other authors have confirmed this lack of activity against food bacteria [45], antimicrobial activity against fitopathogenic bacteria was reported [46]. This compound is produced and released by some plants only as a natural defense in challenging environments.

Pure ursolic acid had inhibitory and bactericidal activity against the three gram-positive bacteria, *L. monocytogenes*, *E. faecium* and *S. aureus*; however, while *L. monocytogenes* and *E. faecium* MIC and MBC values ranged from 33 to 66 μg/mL, *S. aureus* was more resistant, with MIC and MBC values up to 263 μg/mL.

*L. luisieri* maceration extracts, PV and DV, were also active in inhibiting these bacteria. The lack of activity against gram-negative strains could be a consequence of the restricted penetration because of their different and more complex cell walls, since they have a second external phospholipid bilayer designed to reduce permeability to all compounds [47].

The initial maceration extract was only active against *L. monocytogenes* and *E. faecium*, MIC 286–286 μg/mL and MBC 557–1146 μg/mL, respectively. The corresponding ursolic acid concentration in MIC and MBC was 11.3 μg/mL and 45.4–59.6 μg/mL.

After the fractionation process, the PV fraction had better antimicrobial results than ME results. The precipitated solid showed activity against the three gram-positive bacteria. The MIC and MBC values against *L. monocytogenes* (242–83 μg/mL) and *E. faecium* (242–242 μg/mL) were lower than those obtained with the maceration extract and also had inhibitory properties against *S. aureus* (1933 μg/mL). The corresponding ursolic acid concentration in the PV inhibitory and bactericidal values ranged from 28.6 to 56.9 μg/mL. In this case, the final terpene concentration in PV MIC (UA 28.7–57.3 µg/mL) was lower compared with the inhibitory and biocidal concentrations of the pure active (33–66 µg/mL). Although this compound is concentrated in PV regarding the initial maceration extract, the activity cannot entirely be attributed to it. Nevertheless, oleanolic acid, the other triterpene in this work, with a very similar molecular structure, may contribute to the final activity by addition or synergistic effect. Indeed, several studies have reported the antimicrobial activity of oleanolic and ursolic acids. There is an inhibitory capacity against both gram-negative and gram-positive and a synergetic effect between ursolic acid and aminoglucoside antibiotics [48] when tested against 12 bacterial strains. Indeed, oleanolic and ursolic acids have antimicrobial activity which has been related to the peptidoglycan structure, bacterial gene expression and biofilm formation [23].

These differences between gram-positive and gram-negative bacteria was observed by Lai et al., [15] who performed a methanolic extraction of the aerial parts of *L. luisieri.* In this work, the extract was dissolved in DMSO 100%, which may affect the strain sensitivity to the extracts [38].

Even though rosmarinic acid did not show activity on its own, its combination with other non-identified compounds could enhance it. In any case, the concentrations required to inhibit and kill bacterial strains by *L. luisieri* maceration and supercritical extracts were higher than those obtained with the essential oil. This decreased antimicrobial activity could be a consequence of the diverse compounds that constitute these extracts. Their penetration into bacterial cells and action mechanism to inactivate them could be different. Some authors studying new treatments for food preservation have observed a synergistic effect between heat treatment in combination with the addition of natural plant extracts, allowing the reduction of the heat conditions [27].

### 3.4. Antioxidant Activity 

The antioxidant activity was evaluated against the free radical DPPH. Results represented in Figure 3 showed that for every compound and extract, the activity was concentration dependent. *L. luisieri* ethanolic maceration extract and its supercritical fractions showed antioxidant activity, but always lower than the positive controls trolox (IC_50_ 3.5 ± 0.3 µg/mL) and rosmarinic acid (IC_50_ 1.7 ± 0.1 µg/mL). The antioxidant measure of *L. luisieri* maceration extract IC_50_ was 30.66 ± 1.9 µg/mL. This result is similar to other published results about different *Lavandula* species and extracts. The DPPH scavenging activity of the methanolic extract of *L. stoechas* was also studied previously [49]; its IC_50_ value was 34.2 ± 3.1 µg/mL, which was related to its phenolic acid content, 25.2 ± 0.4 mg GAE/g.

Other authors reported free radical inhibition of other *Lavandula* spp extracts. *L. x intermedia* ethyl acetate and ethanolic extracts reported results were IC_50_ = 50.4 µg/mL and IC_50_ = 15.1–45.3 µg/mL, respectively. *L. angustifolia* ethanolic extract IC_50_ was 10.6–33.9 µg/mL; ethanolic extracts from *L. coronopifolia* and *L. multifida* IC_50_ = 15.8 µg/mL and IC_50_ = 19.3 µg/mL, respectively [49,50,51]. In contrast, other authors [52] reported for the *L. stoechas* methanolic extract an IC_50_ of 300 ± 10 µg/mL. The decreased antioxidant activity reported could be a consequence of the heat application in the extraction process and the different proportion of extract: DPPH applied. 

After the supercritical processing of maceration extract, the antioxidant activity showed an increase in PV, IC_50_ to 16.17 ± 0.7 µg/mL. The DV fraction, however, was the less active, not inhibiting the 50% of DPPH free radicals in the studied range of concentrations. These results can be graphically observed in Figure 3, where the PV and DV antioxidant curves are in the left and right, respectively, of the maceration extract curve.

According to these results and the chemical analysis, after the supercritical fluid process, the antioxidant compounds extracted in maceration extract were mainly concentrated into the solid of PV fraction. The increased scavenging activity of PV regarding the initial maceration extract and the lack of activity in DV may be a consequence of rosmarinic acid, which completely precipitates during the supercritical process when converging with CO_2_, increasing its proportion in this fraction from 51.5 mg/g of maceration extract to 93.5 mg/g of PV. The IC_50_ of pure rosmarinic acid was 1.7 µg/mL, which correlates with the rosmarinic acid concentration in the maceration extract and PV IC_50_, 1.5 µg/mL. 

On the other hand, oleanolic and ursolic acids did not seem to scavenge DPPH free radicals since the DPPH inhibition (%) did not reach 50% at the highest concentration tested, even though their concentration in the second fraction was 48.7 mg/g and 56.1 mg/g, respectively. This result did not correspond to other authors, who reported that ursolic acid IC_50_ was 59.70 µg/mL [48]. 

*L. luisieri* scavenging activity seems to be related to its content in rosmarinic and not in oleanolic and ursolic acids. Other *Lavandula* species extracts have also been reported to display several antioxidant mechanisms such as reductive potential, organic, cation and superoxide free radical scavenging, electron or metal cation chelation [52,53,54,55] and has always been related to their content in phenolic compounds such protocatechuic acid, caffeic acid and rosmarinic acid.

## 4. Conclusions

*L. luisieri* extracts showed antioxidant activity against the free radical DPPH and antimicrobial activity against *Listeria monocytogenes*, *Enterococcus faecium* and *Staphylococcus aureus*, and two gram- negative bacteria, *Salmonella* Typhimurium and *Escherichia coli*. Although the essential oil showed a strong antimicrobial activity against the studied bacterial strains, its application could be limited because of its organoleptic properties, which should be further evaluated. The application of the Supercritical antisolvent green technology to the non-volatile fraction of *L. luisier*, not only allowed the concentration of bioactive compounds in a final solid product but also the enhancement of the studied bioactivities in the precipitated solid fraction. The increased antimicrobial activity seems to be a consequence of the ursolic acid enrichment and the antioxidant activity because of rosmarinic acid. Nevertheless, a thorough analysis by HPLC-MS to know the full chemical composition of the ethanolic extract and its supercritical fraction could be interesting for identifying other bioactive compounds. This improved final product could have potential applications in food, cosmetic and pharmaceutical industries as a preservative or nutraceutical. However, it is recommended to perform additional studies in order to assess the preservative effect after its application in the final product as well as its safety for consumers.

## Figures and Tables

**Figure 1 plants-08-00455-f001:**
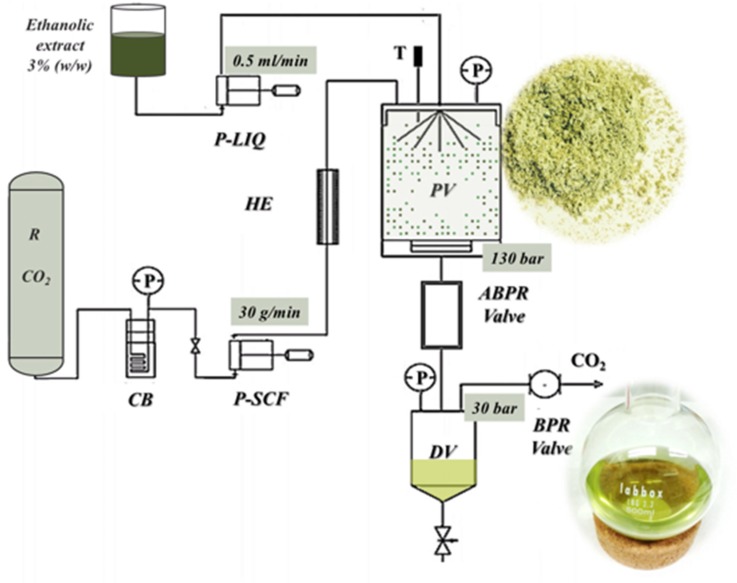
Scheme of the SAF plant. Feed solution reservoir (FS); liquid pump (P-LIQ); CO_2_ reservoir (R), cooling bath (CB); CO_2_ pump (P-SCF); heat exchanger (HE); precipitation vessel (PV); Thermopar (T); automated back pressure regulator (ABPR); back pressure regulator (BPR); downstream vessel (DV).

**Figure 2 plants-08-00455-f002:**
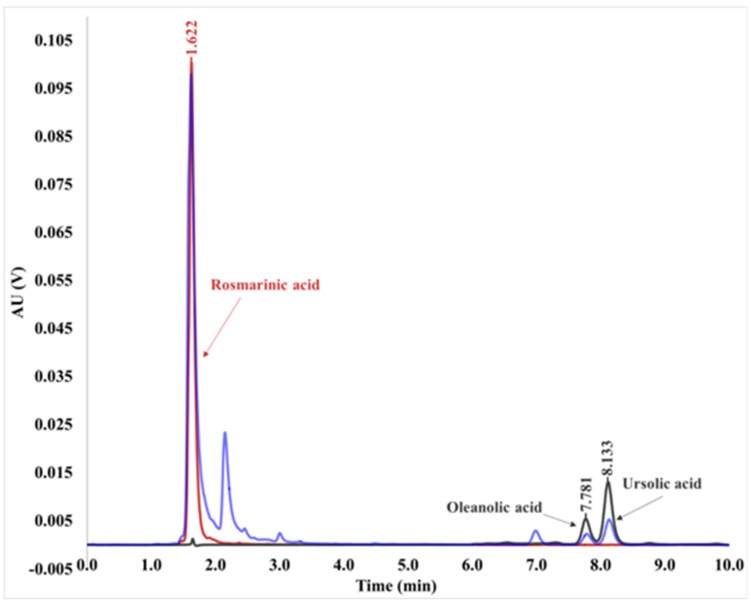
Overlayed chromatograms at 330 nm (0–6 min) and 210 nm (6–10 min) of pure rosmarinic acid (red, retention time 1.622 min), pure oleanolic and ursolic acids (black, retention times 7.781 min and 8.133 min) and the ethanolic extract of *L. luisieri* (blue).

**Figure 3 plants-08-00455-f003:**
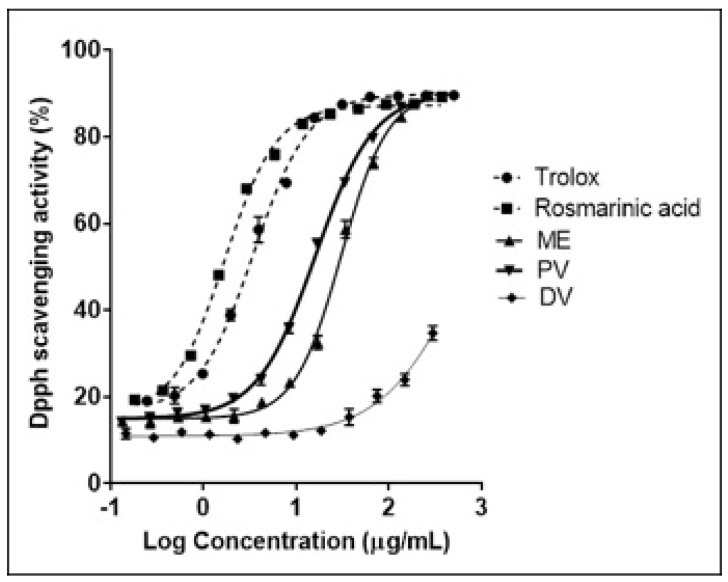
Logarithmic curve representation of the antioxidant activity of *Lavandula luisieri* ethanolic maceration extract (ME) and its SAF fractions: precipitation vessel fraction (PV) and downstream vessel fraction (DV). Positive controls 6-hdroxy-2,5,7,8-tetramethylchromane- 2-carboxylic acid (Trolox) and rosmarinic acid (RA).

**Table 1 plants-08-00455-t001:** Rosmarinic, oleanolic and ursolic acids mg quantified in the maceration extract and each supercritical fraction.

	RA (mg/g)	OA (mg/g)	UA (mg/g)
ME	51.4 ± 0.6	28.8 ± 2.3	52.0 ±1.4
PV	93.5 ±0.5	26.4 ± 1.8	118.6 ± 1.1
DV	-	48.7 ± 1.8	56.1 ± 1.1

ME: Maceration Extract; PV: Precipitation Vessel; DV: Downstream vessel.

**Table 2 plants-08-00455-t002:** *Lavandula luisieri* essential antimicrobial results against bacterial strains. Inhibition zone (mm) and minimum inhibitory and bactericidal concentrations (μL/mL).

Strain	Inhibition Zone (mm)	MIC (μL/mL)	MBC (μL/mL)
*L. monocytogenes*	+++	0.5	3
*E. faecium*	+	0.5	1
*S. aureus*	+++	0.5	0.5
*S.* Typhimurium	+++	5	5
*E. coli*	+	30	-

(-) not halo (+) non-inhibitory (++) moderately inhibitory (+++) strongly inhibitory.

**Table 3 plants-08-00455-t003:** Minimum inhibitory and bactericidal concentrations (μL/mL) of usolic acid, *L. luisieri* ethanolic maceration extract (ME), and the supercritical fractions PV and DV against *L. monocytogenes*, *E. faecium* and *S. aureus*.

Extract or Pure Active	*L. monocytogenes*		*E. faecium*		*S. aureus*	
	MIC	MBC	MIC	MBC	MIC	MBC
**Ursolic acid**	33	66	66	66	263	263
**ME**	286	557	286	1146	-	-
**PV**	242	483	242	242	1933	-
**DV**	232	931	-	-	-	-

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
