# Peer review of "Supercritical Carbon Dioxide Antisolvent Fractionation for the Sustainable Concentration of Lavandula luisieri (Rozeira) Riv.- Mart Antimicrobial and Antioxidant Compounds and Comparison with Its Conventional Extracts"

_plants, 2019, doi:10.3390/plants8110455_

Round 1
Reviewer 1 Report
the manuscript is well written. i only would like to suggest to revise the conclusion and make more details about limitations and possible future work.
Author Response
Review Report (Round 1)
Comments and Suggestions for Authors
The manuscript is well written. I only would like to suggest to revise the conclusion and make more details about limitations and possible future work.
According to the reviewer’s comments, limitations regarding organoleptic effects of the essential oils have been emphasized. In addition, it has been mentioned the possible future work to be done.
Reviewer 2 Report
Please check the attached file.

Author Response
Reviewer 2
Review Report (Round 1)
Comments and Suggestions for authors
Overall a beautiful article: readable and interesting due to its applied approach (antimicrobial). Yet, I would have liked to see some chromatograms from HPLC especially because the authors say they quantified rosmarinic, oleanolic and ursolic acids and it would have been interesting to see the peaks in fractions compared to pure standards.
All the minor changes have been introduced in the new version of the manuscript.
Regarding chromatographs we have decided not to published them not to centre the attention in the analytical methodology but only in the results and their consequences ,related to the increased of the antimicrobial and antioxidant activities as a consequence of the application of the Supercritical antisolvent green technology that allowed for the concentration of bioactive compounds.
Minor changes should be done before publishing, regarding English language phrasing, grammar, typos, like:
Line 19: correctly would be: “Lavandula stoechas subsp. luisieri (Rozeira) Rozeira is a Spanish subspecies from..”. I would say “its essential oil..” or “essential oil extracted has been..” (without which). It has been corrected according to the reviewer’s comment.
Line 20: there is another font and/or size used at the end of the phrase. Corrected
Line 23: comma before “respectively”. Comma has been added as suggested.
Line 24: instead or “was processed” ‐> “by means of”/”using”. It has been changed.
Line 25: after “rosmanic acid” ‐> delete comma. Comma has been deleted
Line 26: instead of “quantified”, I would recommend “evaluated”. The word “evaluated” has been used instead of “quantified”
Line 27: check font for bacteria. The same font has been applied.
Line 28: The phrase should flow and underline the subjects, so I’d say: “Antimicrobial activities of all extracts and pure identified compounds (a &b) were evaluated against five bacterial strains (x, y, z etc.). The phrase has been changed as indicated
Line 30: before “the polyphenol rosmarinic..” ‐> comma. Corrected
Line 31: instead of “regarding”‐> compared to. So it has been modified
Line 33: instead of “for its” ‐> “with potential application”. It has been changed as indicated.
Line 61: …” information on extracts is scarce” + separate “sofar” + “only two other studies have interest in its extracts but high temperature was applied”. It has been changed as indicated.
Line 69: delete “in any case”. The words “In any case” have been delated
Line 72: delete comma before “and”. Corrected
Line 80: delete “nevertheless” and comma. Both have been deleted
Line 90: delete “because…precedents”. It has been changed as indicated.
Line 153: compounds names with small letters. Compound names have been written in small letters.
Line 171: instead of “planting” ‐> performing 1:10 dilutions…/diluting 1:10 in… It has been corrected.
Line 182: space before reference. The space has been included
Line 199: it is advisable (not mandatory) to present values in C% (v:v) ‐> throughout the text. In order to do more comparable our results with other studies we have expressed the concentrations assayed as usual in these kind of scientific works.
Line 220: why would the authors express both in ug and ul? Please evaluate your procedure so that to be uniform (essential oils and their compounds can be expressed both ways, but it is advisable to use only one). It has been expressed in such a way because the essential oil density is unknown. Thus, essential oils concentrations assayed were given in volume/volume, while the extracts were weighed and so the units are given in mass/volume.
Line 246: data in parenthesis is redundant. At present, data are not shown
Line 249 ‐ 250: “..of actives but it can cause…compounds; neither does it provide better extraction yield.” It has been revised
Line 258 ‐ 259: rephrase, for instance: “in the composition of …only polyphenol,
…were identified by means of HPLC” It has been rephrased
Line 260: put the ref in parenthesis. So it has been done
Line 267: “1.9 times more concentrated compared to the initial ME” Corrected as suggested
Line 284: bacteria susceptibility. The phrase has been rewritten
Line 285: rephrase smth like “E. faecium (x mm) and E coli (y mm) showed no inhibition when EO was used at (concentration). Corrected
Line 287: L monocytogenes was the most susceptible bacterium (x mm at C = y”/a ug/ml) followed by…. Corrected as indicated
Line 288: delete “in these cases..activity”. The whole phrase has been eliminated.
Line 290: methodology goes where it is its place (for quantitative..etc.). Those references related to the methodology section have been deleted
Line 294: delete comma after “obtained” + delete space after parenthesis + “whereas higher MIC valuers..” It has been corrected as indicated.
Line 298: space between value and ul. Space has been eliminated
Line 306: check reference number typo. Corrected
Line 311: instead of “being” ‐> “with”. Corrected
Line 332: “with other substances like fenchone…” Modified as indicated.
Line 343: delete “also” . “Also" has been deleted
Line 249: instead of the redundancy and irrelevant info, use smth like “Salmonella and E.coli did no…concentrations (table xx)” ‐> delete the rest. Corrections have been included in the text
Line 353: delete “regarding…assayed2 ‐> “Rosmarinic acid…” Corrected.
Line 355: after comma ‐> antimicrobial activity against phytpathogenic bacteria was Reported. Corrected as indicated
Line 357: “is released only as a natural defence in challenging environments” The phrase has been rewritten.
Line 385: check typo in reference Corrected
Line 383‐388: the reader does not understand what you argue; rephrase or delete if “is not comparable”.. It has been rewritten in order to be better understood.
Line 390: which other compound/compounds?. We refer to other compounds not identified yet. We have indicated it in the text
Line 412: the decreased…could be a consequence… Corrected as indicated
Reviewer 3 Report
I think that the authors deal with an interesting subject, whose approach and experimental design are suitable for me. Therefore, it could be a good article to be published in this journal. Moreover, I do not have any "major requirements" about this research work.
However, some "minor revision" is required, as follows:
Concerning the abstract, too little information about the usefulness o this study is done. It is mainly focused in the experimental procedure. Thus, rewrite it to show some background, the aim of the work, the procedure and the conclusions in a more balanced way. In addition, there are different font styles and sizes. Keywords: In order to improve the "scope" of your article, when published, you should avoid repeating the same words than in the title. Thus, remove "Supercritical antisolvent fractionation" and replace it for another keyword. Line 107: Where does the lavender from Toledo come from, exactly? Another research centre? Local growers? Explain how you got these plants or seeds. As you do not have many figures or tables, I encourage you to add a table or figure to explain the chemical composition of the essential oil, instead of putting the reference only (line 117). Line 120: Explain how you removed the hexane from the first maceration. The format of the text should be improved. There are many "typos", such as "sofar" (line 61), "et al" without the dot (line 64), "SigmaAldrich" in line 97, etc. There are different citation formats in lines 89, 301, 324, 385, as far as I can tell. Line 334, the citation order is not right. The English, also, should be revised: I recommend to replace which for whose in lines 19 and 46, for instance. "Works" in line 62. 6-hYdroxY"... in line 104. Whereas instead of while in line 154. Replace "y" for "and" in line 314. Acetate in line 320. And there is an English mistake that I usually made and many reviewers corrected me, and now it is my turn to correct you. The structure used in line 311 (being + subject), that is widely used in Spanish-speaking countries, is not common in English. Therefore, rewrite the sentence avoiding this structure, because it will be confusing for many readers.
Author Response
Reviewer 3
Comments and Suggestions for authors
I think that the authors deal with an interesting subject, whose approach and experimental design are suitable for me. Therefore, it could be a good article to be published in this journal. Moreover, I do not have any "major requirements" about this research work.
However, some "minor revision" is required, as follows:
Concerning the abstract, too little information about the usefulness o this study is done. It is mainly focused in the experimental procedure. Thus, rewrite it to show some background, the aim of the work, the procedure and the conclusions in a more balanced way. In addition, there are different font styles and sizes. The abstract has been rewritten in order to come across the reviewer’s requirements.
Keywords: In order to improve the "scope" of your article, when published, you should avoid repeating the same words than in the title. Thus, remove "Supercritical antisolvent fractionation" and replace it for another keyword. The acronym has been used instead.
Line 107: Where does the lavender from Toledo come from, exactly? Another research centre? Local growers? Explain how you got these plants or seeds. As you do not have many figures or tables, I encourage you to add a table or figure to explain the chemical composition of the essential oil, instead of putting the reference only (line 117).
Clarified. Lavandula luisieri plant material was collected from Cariñena, Zaragoza (Spain) and provided by the Agrifood Research and Technology Centre of Aragon (CITA, Spain). This plant was adapted to the Cariñena experimental field in 2008 from a Toledo (Spain) wild population. It has been better indicated in the text. Despite of being very interesting to know the chemical composition of the essential oil, these are data coming from a previous work that have been already published and so it should not be shown again in another work.
Line 120: Explain how you removed the hexane from the first maceration. The format of the text should be improved. Clarified
There are many "typos", such as "sofar" (line 61), "et al" without the dot (line 64), "SigmaAldrich" in line 97, etc. There are different citation formats in lines 89, 301, 324, 385, as far as I can tell. Line 334, the citation order is not right. We have tried to correct these “typos” as long as we have detected them.
The English, also, should be revised: I recommend to replace which for whose in lines 19 and 46, for instance. The latter has been modified according to the reviewer’s indication but the former had been changed before according to the previous reviewer’s comments. "Works" in line 62. This work has been substituted by “studies”. 6-hYdroxY"... in line 104. It has been corrected as indicated
Whereas instead of while in line 154. Replace "y" for "and" in line 314. Acetate in line 320. And there is an English mistake that I usually made and many reviewers corrected me, and now it is my turn to correct you. The structure used in line 311 (being + subject), that is widely used in Spanish-speaking countries, is not common in English. Therefore, rewrite the sentence avoiding this structure, because it will be confusing for many readers. All the indications given have been corrected in the text